# Auxin exposure disrupts feeding behavior and fatty acid metabolism in adult *Drosophila*

Sophie A Fleck[1], Puja Biswas[2], Emily D DeWitt[1], Rebecca L Knuteson[1], Robert C Eisman[1], Travis Nemkov[3], Angelo D'Alessandro[3], Jason M Tennessen[1], Elizabeth Rideout[2], Lesley N Weaver[1]*

[1]Department of Biology, Indiana University, Bloomington, United States; [2]Department of Cellular and Physiological Sciences, Life Sciences Institute, The University of British Columbia, Vancouver, Canada; [3]Department of Biochemistry and Molecular Genetics, Anschutz Medical Campus, University of Colorado School of Medicine, Aurora, United States

**Abstract** The ease of genetic manipulation in *Drosophila melanogaster* using the *Gal4/UAS* system has been beneficial in addressing key biological questions. Current modifications of this methodology to temporally induce transgene expression require temperature changes or exposure to exogenous compounds, both of which have been shown to have detrimental effects on physiological processes. The recently described auxin-inducible gene expression system (AGES) utilizes the plant hormone auxin to induce transgene expression and is proposed to be the least toxic compound for genetic manipulation, with no obvious effects on *Drosophila* development and survival in one wild-type strain. Here, we show that auxin delays larval development in another widely used fly strain, and that short- and long-term auxin exposure in adult *Drosophila* induces observable changes in physiology and feeding behavior. We further reveal a dosage response to adult survival upon auxin exposure, and that the recommended auxin concentration for AGES alters feeding activity. Furthermore, auxin-fed male and female flies exhibit a significant decrease in triglyceride levels and display altered transcription of fatty acid metabolism genes. Although fatty acid metabolism is disrupted, auxin does not significantly impact adult female fecundity or progeny survival, suggesting AGES may be an ideal methodology for studying limited biological processes. These results emphasize that experiments using temporal binary systems must be carefully designed and controlled to avoid confounding effects and misinterpretation of results.

*For correspondence:
lnweaver@iu.edu

Competing interest: The authors declare that no competing interests exist.

## eLife assessment

This **valuable** study shows that auxin exposure perturbs feeding behavior, survival rates, lipid metabolism, and gene expression patterns in adult *Drosophila* flies. The results are **solid** with proper methods and data analyses, and the evidence broadly supports the conclusions with only minor weaknesses. This work is relevant for fly geneticists who are interested in using the auxin-inducible gene expression system for inducing target protein degradation acutely.

## Introduction

The intricate dissection of cell-type-specific processes in *Drosophila* is largely dependent on the yeast-derived *Gal4/UAS* binary system, which allows manipulation of biological pathways in a spatial and temporal manner (**Brand and Perrimon, 1993**). This methodology utilizes the Gal4 transcription

factor that is under control of a tissue-specific promoter to induce transgene expression downstream of an Upstream Activating Sequence (UAS). Conditional control of gene expression using changes in temperature (*McGuire et al., 2003*) or feeding of small molecules (*Roman et al., 2001*; *Potter et al., 2010*; *McClure et al., 2022*) restricts Gal4 activity to specific developmental timepoints. For example, use of a temperature sensitive *Gal80* mutant transgene (*Gal80ts*, an inhibitor of Gal4; *Douglas and Hawthorne, 1966*) allows for temporal control of Gal4 activity with a simple shift to the *Gal80ts* restrictive temperature (29°C; *McGuire et al., 2003*). Conversely, drug-inducible systems such as *GeneSwitch* and the *Q-system* control temporal and reversible transgene expression with administration of RU486 or quinic acid, respectively, without the need to rear flies at the *Gal80ts* permissive temperature (18°C). These modifications to the *Gal4/UAS* system have improved the capability to characterize the roles of essential biological pathways in a tissue-specific manner while avoiding lethality at key developmental stages.

Despite these advancements, each methodology has caveats that must be considered. For example, rearing flies containing *Gal80ts* at 18°C nearly doubles the developmental time from egg to adult (*Powsner, 1935*). In addition, the relatively high restrictive temperature needed to inactivate Gal80ts has adverse effects on physiological processes including circadian rhythm (*Parisky et al., 2016*), aging (*Miquel et al., 1976*), and progeny survival (*Gandara and Drummond-Barbosa, 2022*). Similarly, use of RU486 has been demonstrated to repress muscle-specific mitochondrial genes (*Robles-Murguia et al., 2019*) and lipogenesis (*Ma et al., 2021*), among other defects (*Landis et al., 2015*; *Yamada et al., 2017*), making the *GeneSwitch* system non-ideal for certain experiments. Although designed to provide flexibility in experimental design and remove temperature-related defects, the alterations in physiology, behavior, and lifespan [some of which are not shared between the sexes; (*Landis et al., 2015*)] due to RU486 feeding impose difficulties in data interpretation.

The auxin-inducible gene expression system (AGES) was recently developed as an alternative method to induce transgene expression and is compatible with the breadth of *Gal4* transgenic lines available (*McClure et al., 2022*). In this system, *Gal80* is fused to auxin-inducible degron tags that target Gal80 for degradation, allowing for Gal4-mediated transgene induction upon auxin consumption. This system poses substantial advantages. For example, flies can be reared at the optimal temperature for development (25°C) and transgene expression is strictly induced with supplementation of auxin to the media. In addition, both control and experimental animals are genetically identical (similar to the *GeneSwitch* and *Q-system*), thus minimizing differences that may arise due to genetic variation.

However, there is emerging evidence that insects can synthesize auxin (*Yokoyama et al., 2017*; *Tokuda et al., 2022*), suggesting that there may be important biological roles for this hormone in *Drosophila*. While exposure to 10 mM 1-naphthaleneacetic acid (the widely employed synthetic hormone in the auxin family and hereafter referred to as 'auxin') has been reported to have no effect on insect development, survival, or movement in a wild-type *Drosophila* strain (*McClure et al., 2022*), it remains unclear whether auxin exposure affects development in other commonly used *Drosophila* genetic background strains. It is also unknown whether increased auxin exposure in adult *Drosophila* results in subtle defects in physiological processes that may confound experimental interpretations. In this study, we sought to test whether auxin affects larval development in additional strains, and to determine whether auxin exposure in adults leads to defects in metabolism, the transcriptome, and oogenesis. We found that recommended concentrations of auxin for AGES disrupt feeding behavior, whereas increasing levels of auxin results in physiological changes and lethality. Additionally, we found that auxin exposure delays larval development, decreases triglyceride levels, and alters the transcriptomic profile of fatty acid metabolism genes in both sexes. Finally, despite the decrease in circulating lipids, auxin does not severely disrupt processes of oogenesis or progeny survival, suggesting AGES may be an appropriate method to use for some studies. Our results highlight changes in development and physiology that should be considered when using auxin to manipulate gene and protein expression in larval and adult *Drosophila*, as well as other factors researchers should account for in experimental design using temporal control of the *Gal4/UAS* system.

## Results and discussion

### Adult *Drosophila* males have increased sensitivity to auxin compared to females

We sought to utilize the AGES expression system in our laboratory for controlling gene expression and began by determining whether we could recapitulate Gal4 expression at levels like that of *Gal80[ts]*. We recombined the auxin-inducible degron line (*AGES*) with *3.1Lsp2-Gal4* (*3.1Lsp2-Gal4[AGES]*) to drive the expression of *UAS-nucGFP* in adult female adipocytes compared to the previously described *3.1Lsp2-Gal4* recombined with *Gal80[ts]* (*3.1Lsp2Gal4[ts]*; *Armstrong et al., 2014*). Zero- to two-day-old females of each genotype were fed inactive yeast paste for 2 days (pre-treatment) prior to Gal4 induction. Gal4 expression was induced in females carrying the *AGES* transgene by feeding solid food supplemented with inactive wet yeast paste containing 0 mM to the recommended dose of 10 mM auxin (inactive yeast was used to prevent auxin metabolism by live yeast) for 2 days; whereas females carrying the *Gal80[ts]* transgene were shifted from 18 to 29°C for 2 days and the relative expression of *GFP* transcripts were measured (*Figure 1—figure supplement 1A*). There was minimal 'leaky' Gal4 expression with *3.1Lsp2-Gal4[AGES]* fed food without auxin (0 mM); whereas 5 and 10 mM auxin-induced *nucGFP* expression at levels similar to that of *3.1Lsp2-Gal4[ts]*. Furthermore, we found that removal of auxin using *3.1Lsp2-Gal4[AGES]* inactivated Gal4 activity at a faster rate than *3.1Lsp2-Gal4[ts]* (shifted back to 18°C; *Figure 1—figure supplement 1B*). Therefore, Gal4 expression using AGES can be recapitulated as previously described (*McClure et al., 2022*) and is faster at repressing transgene activation compared to *Gal80[ts]*.

Because the wild-type strain *Canton-S* was previously used to monitor development and survival in flies reared on auxin-containing food (*McClure et al., 2022*), we monitored larval development in flies from the *w[1118]* genotype, a widely used genetic background strain in *Drosophila* biology. When we measured the time between egg laying and pupariation in a mixed-sex group of larvae reared on yeast–sugar–cornmeal food supplemented with (5 mM) or without (0 mM) auxin, we found that the time to pupariation was longer in auxin-fed larvae than in larvae raised without auxin (*Figure 1—figure supplement 1C*). This suggests that the recommended dose of auxin delays larval development in at least one widely used *Drosophila* strain, indicating that genetic background is an important consideration when using the AGES system for studies at the larval stage of development.

Given these larval phenotypes, we next wanted to determine whether auxin exposure induces physiological changes in adults. Because genetic variation can influence physiology (*Shorter et al., 2015*; *Evangelou et al., 2019*; *Damschroder et al., 2020*), we used multiple strains to test the effects of auxin on physiology. We first compared the *AGES* line and the strain into which the *AGES* transgene was introduced (*VK00040*) in our experiments. We also used *Oregon-R* [the wild-type strain used for the ModENCODE project; (*Roy et al., 2010*)] as an additional control. We incubated 4- to 6-day-old adult males and females of each genotype in vials supplemented with chromatography paper soaked with liquid food containing 0 to 350 mM auxin and measured lethality after 48 hr of exposure (*Figure 1A–C*; *Figure 1—figure supplement 2A*). Compared to the 0 mM control, both males and females showed increased susceptibility to moderate concentrations of auxin (20–30 mM; *Figure 1B, C*; *Figure 1—figure supplement 2A*). In addition, we observed sex differences in sensitivity to auxin exposure, with females able to tolerate higher concentrations relative to males. For example, the benchmark dose (BMD; the concentration that produces a change in the response) in males was lower than that of females in each genotype (*Table 1*). Interestingly, both *AGES* males and females had a higher BMD upon auxin exposure relative to the *VK00040* injection and *Oregon-R* lines, suggesting that the *AGES* transgene may confer resistance to auxin toxicity. These results suggest that exposure to auxin in both males and females results in lethality, where males have a higher susceptibility to auxin exposure compared to females.

For the remainder of our analyses, we used the recommended working concentration of auxin for AGES (10 mM), which as previously described, did not cause lethality in the *AGES* line for either sex (*Figure 1B, C*; *Table 1*; *McClure et al., 2022*). To determine whether feeding behavior could account for sex differences in the survival response to auxin, we analyzed the food consumption of adult male and female flies exposed to 0 or 10 mM auxin using the Fly Liquid-Food Interaction Counter (FLIC; *Ro et al., 2014*) over a 24-hr period (*Figure 1D–G*, *Figure 1—figure supplement 2B–E*). In both feeding conditions, females exhibited more feeding events relative to males in all tested genotypes (*Table 2*). For example, females fed 0 mM auxin in both the *VK00040* and *AGES* lines had significantly

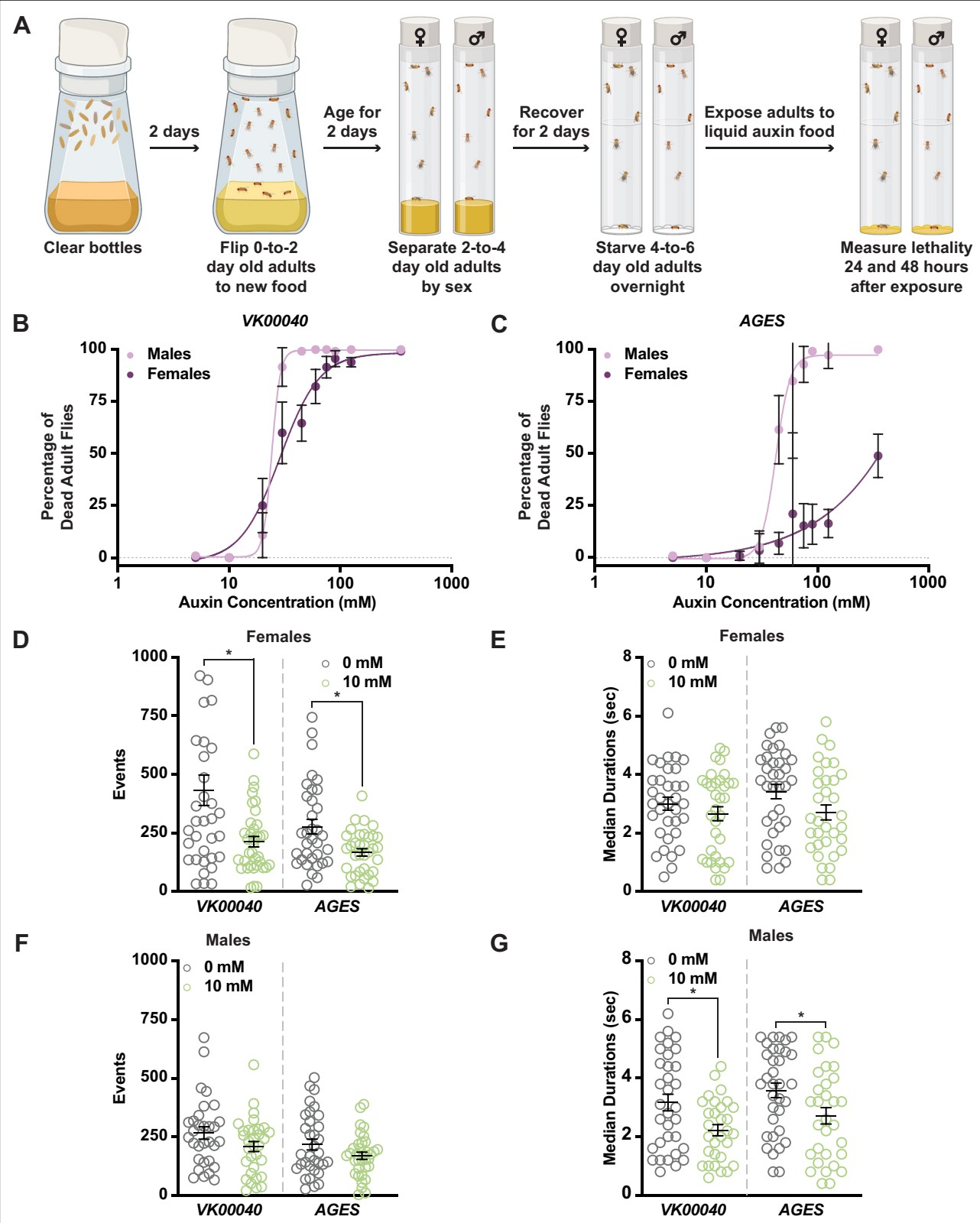

**Figure 1.** Increased auxin exposure in adult *Drosophila* decreases survival and alters feeding behavior. (**A**) Cartoon schematic of auxin exposure feeding protocol. Cartoon was created using BioRender. Dose–response curves for males and females exposed to increasing concentrations of auxin in the *VK00040* transgene injection line (**B**) or the *AGES* transgenic line (**C**). Twenty flies per sex and genotype were analyzed. Data shown as mean ± standard deviation (SD). (**D**) The total number of feeding events for females in the *VK00040* or *AGES* fly line exposed to 0 or 10 mM auxin. (**E**) The median time

*Figure 1 continued on next page*

*Figure 1 continued*

of feeding activity of females in the *VK00040* or *AGES* fly lines exposed to 0 or 10 mM auxin. (**F**) The total number of events for males in the *VK00040* or *AGES* fly lines exposed to 0 or 10 mM auxin. (**G**) The median time of feeding activity of males in the *VK00040* or *AGES* fly lines exposed to 0 or 10 mM auxin. At least 30 flies per sex were analyzed. Data shown as mean ± standard error of the mean (SEM). *p < 0.05, Mann–Whitney *U*-test.

The online version of this article includes the following source data and figure supplement(s) for figure 1:

**Source data 1.** Lethality analysis for *VK00040* males and females.

**Source data 2.** Lethality analysis for *AGES* males and females.

**Source data 3.** FLIC data for *VK00040* and *AGES* females.

**Source data 4.** FLIC data for *VK00040* and *AGES* males.

**Figure supplement 1.** Auxin exposure increases the time of development.

**Figure supplement 1—source data 1.** Analysis of Gal4 expression in *3.1Lsp2-Gal4*[ts] and *3.1Lsp2-Gal4*[AGES] females.

**Figure supplement 1—source data 2.** Analysis of Gal4 expression in *3.1Lsp2-Gal4*[ts] and *3.1Lsp2-Gal4*[AGES] females over time.

**Figure supplement 1—source data 3.** Larval developmental time in the presence or absence of auxin.

**Figure supplement 2.** Auxin exposure in *Oregon-R* adults increases lethality and alters female feeding behavior.

**Figure supplement 2—source data 1.** Lethality analysis for *Oregon-R* males and females.

**Figure supplement 2—source data 2.** FLIC analysis for *Oregon-R* females.

**Figure supplement 2—source data 3.** FLIC analysis for *Oregon-R* males.

more engagements with the interaction counter compared to *VK00040* and *AGES* males (compare *Figure 1D* to *Figure 1F*). In addition, genetic background confers differences in feeding behavior as seen by the difference between *VK00040* and *AGES*. Surprisingly, compared to the 0 mM food, females had fewer interactions and slightly decreased event durations when fed the 10 mM auxin food; whereas males only decreased the amount of time they interacted with the food when exposed to 10 mM auxin (*Figure 1G*). These results are consistent with previous reports that females eat more than males on normal food diets with yeast (*Wong et al., 2009*). However, our results also suggest that females exposed to auxin may have an increased survival due to fewer interactions with the food relative to adult males, resulting in differential auxin sensitivity. Therefore, future studies should investigate the female-specific avoidance of auxin, since decreased feeding behavior could confound interpretation of results.

## Fatty acid metabolism is decreased in adults exposed to auxin

To determine whether auxin exposure altered metabolism in adult *Drosophila*, we performed metabolomics to compare adult males and females of each genotype exposed to 10 mM auxin relative to the 0 mM control (*Figure 2—source data 1*). Partial least squares discriminant analysis showed that each genotype clustered in distinct groups (*Figure 2—figure supplement 1*). Notably, 0 mM controls clustered separately from 10 mM auxin samples in all genotypes for both sexes along the Component 1 axis, which describes 33.4% or 34.4% of the variance for females and males, respectively. We further analyzed the data using MetaboAnalyst (*Pang et al., 2021*), which revealed that the levels of acylcarnitine and fatty acid metabolites were the most significantly decreased metabolites in females exposed to 10 mM auxin regardless of genotype (*Figure 2A*; *Figure 2—figure supplements 2–4*). Adult males exposed to 10 mM auxin also exhibited decreased fatty acid and acylcarnitine metabolites (*Figure 2B*); however, amino acid levels were the most significantly altered metabolites in males in response to auxin (*Figure 2—figure supplements 2–4*).

To confirm that fatty acid metabolism is altered in response to auxin, we exposed adult males and females of each genotype to 0 or 10 mM auxin and measured the levels of triacylglycerol (TAG) after

**Table 1.** Benchmark dose of male and female *Drosophila* exposed to auxin.

| Genotype | Male | Female |
| --- | --- | --- |
| *Oregon-R* | 8.89 mM | 19.97 mM |
| *VK00040* | 17.446 mM | 14.208 mM |
| *AGES* | 32.449 mM | 57.291 mM |

**Table 2.** Average feeding events and durations in male and female *Drosophila*.

| Genotype | Male | | Female | |
|---|---|---|---|---|
| | 0 mM | 10 mM | 0 mM | 10 mM |
| *Oregon-R* | 342.8 ± 39.85 events<br>3.21 ± 0.26 s duration | 278.72 ± 26.64 events<br>3.21 ± 0.3 s duration | 519.14 ± 81.52 events<br>3.22 ± 0.24 s duration | 268.82 ± 28.65 events<br>3.3 ± 0.29 s duration |
| *VK00040* | 266.42 ± 26.27 events<br>3.17 ± 0.29 s duration | 208.56 ± 21.15 events<br>2.22 ± 0.19 s duration | 432.23 ± 64.82 events<br>3.0 ± 0.22 s duration | 213.03 ± 22.54 events<br>2.66 ± 0.24 s duration |
| *AGES* | 217.15 ± 23.09 events<br>3.58 ± 0.25 s duration | 169.63 ± 15.25 events<br>2.71 ± 0.28 s duration | 277.44 ± 31.19 events<br>3.42 ± 0.25 s duration | 167.49 ± 15.9 events<br>2.71 ± 0.26 s duration |

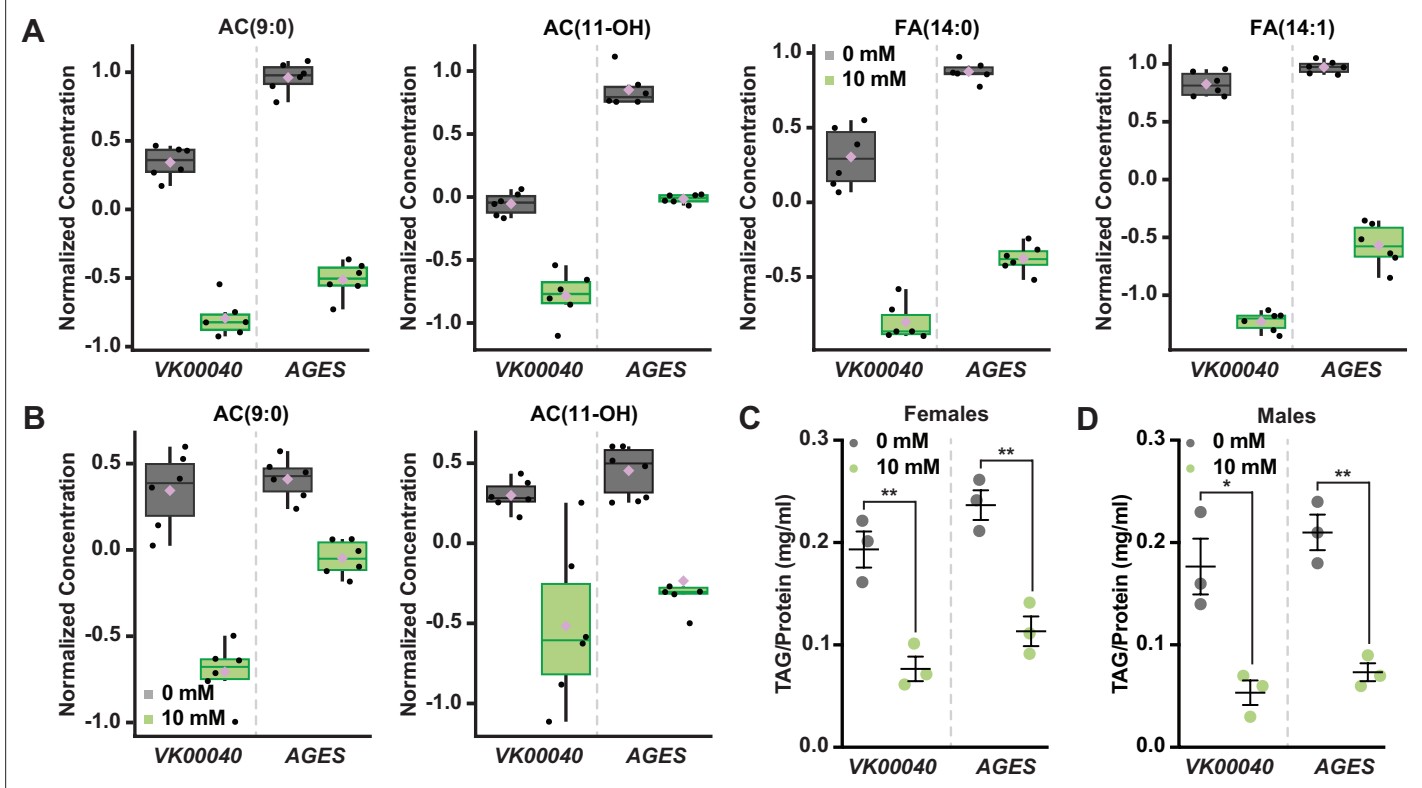

**Figure 2.** Auxin exposure decreases acylcarnitine, fatty acid, and triglyceride levels in adult *Drosophila*. Box plots illustrating the relative abundance of acylcarnitine or fatty acid metabolites in adult females (**A**) or males (**B**) exposed to 0 or 10 mM auxin. All box plots were generated using MetaboAnalyst 5.0 as described in the methods. Black dots represent individual samples, the horizontal bar in the middle represents the median, and the purple diamond represents the mean concentration. For all box plots, the metabolite fold change was >twofold and $p < 0.05$. Triacylglycerol (TAG) contents (mg TAG/mg protein) in whole adult females (**C**) or males (**D**) in the *VK00040* or *AGES* fly lines exposed to 0 or 10 mM auxin. Data shown as mean ± standard error of the mean (SEM). *$p < 0.05$, **$p < 0.01$, two-tailed Student's *t*-test.

The online version of this article includes the following source data and figure supplement(s) for figure 2:

**Source data 1.** Metabolomics raw data for *Oregon-R*, *VK00040*, and *AGES* males and females.

**Source data 2.** Triacylglycerol (TAG) analysis for *Oregon-R*, *VK00040*, and *AGES* males and females.

**Figure supplement 1.** Auxin exposure alters metabolism in adult *Drosophila* males and females.

**Figure supplement 2.** Acylcarnitine and fatty acid levels decrease in *Oregon-R* adult males and females exposed to auxin.

**Figure supplement 2—source data 1.** Body fat analysis for $w^{1118}$ males and females.

**Figure supplement 2—source data 2.** Body fat analysis for $w^{1118}$ males and females after auxin withdrawal.

**Figure supplement 3.** Auxin exposure decreases acylcarnitine and fatty acid metabolites in *VK00040* adult males and females exposed to auxin.

**Figure supplement 4.** Acylcarnitine and fatty acid levels decrease in *AGES* adult males and females exposed to auxin.

48 hr of exposure. Both males and females exposed to 10 mM auxin of each genotype had significantly less TAG relative to the 0 mM control (*Figure 2C, D*; *Figure 2—figure supplement 2G*). We note that the lack of sex difference in lipid levels is likely due to exposing adults to auxin prior to the onset of male–female differences in TAG (*Wat et al., 2020*). To determine whether auxin influences TAG in additional contexts, we transferred newly eclosed virgin $w^{1118}$ males and females to a yeast–sugar–cornmeal diet supplemented with either 0 mM or the recommended dose of 10 mM auxin and measured TAG levels after 5 days. We found a significant decrease in whole-body fat storage in both male and female flies exposed to 10 mM auxin compared with flies transferred to food with 0 mM auxin (*Figure 2—figure supplement 2H*); however, the magnitude of the auxin-induced decrease in body fat was greater in females than in males (sex:diet interaction p = 0.001; two-way analysis of variance [ANOVA]). This suggests that auxin has a stronger effect on female TAG levels, in line with our data showing a female-biased reduction in food interactions on auxin-supplemented medium.

We next asked whether this effect on TAG levels was reversible. We exposed 0-day-old male and female flies to diets with 0 or 10 mM auxin for 5 days, and then measured TAG levels after shifting the flies to food supplemented with no auxin for five additional days. We found that females but not males showed a strong trend toward recovery of whole-body TAG levels after shifting the flies back to food with no auxin (*Figure 2—figure supplement 2I*), suggesting the effect of auxin feeding on body fat were reversible only in females. Furthermore, we found that flies exposed to food supplemented with 10 mM auxin for 10 days show no additional reduction in whole-body TAG levels compared with flies exposed to auxin for 5 days in either sex (*Figure 2—figure supplement 2I*). Taken together, these results suggest that auxin exposure disrupts fatty acid metabolism, resulting in decreased circulating lipids in adult males and females. While these changes to fat metabolism are partially reversible after withdrawal of auxin supplementation in females, changes to whole-body TAG levels persisted even after auxin withdrawal in males.

## Auxin exposure induces global transcriptomic changes in adult *Drosophila*

To determine whether auxin exposure-induced changes in gene expression, we exposed adult males and females of each genotype to 0 and 10 mM auxin for 48 hr and performed RNA sequencing analysis of whole animals. In both males and females of each tested genotype, at least 150 transcripts were significantly altered in response to auxin exposure (*Figure 3*; *Figure 3—figure supplements 1 and 2*). Genes involved in drug metabolism (e.g., glutathione-*S*-transferases and uridine diphosphate-glucuronosyltransferases, phase II enzymes required to increase hydrophobicity of compounds; *Yu, 2008*) were significantly upregulated in response to auxin exposure in both males and females of all genotypes (*Figure 3E*; *Figure 3—figure supplements 1 and 2*), suggesting that auxin induces a xenobiotic response at this concentration (*Yu, 2008*). Surprisingly, genes involved in fatty acid metabolism (e.g., Fad2, which encodes a desaturase) were also significantly upregulated; whereas lipases (enzymes that break down fatty acids) were significantly downregulated in each genotype and sex (*Figure 3F*; *Figure 3—figure supplement 1D*; *Figure 3—figure supplement 2C, D*). Notably, the transcription of enzymes associated with lipolysis such as the lipase *brummer* (*Grönke et al., 2005*) or the perilipins *Lsd-1* and *Lsd-2* (*Beller et al., 2010*) were unaltered in response to auxin exposure (*Figure 3—source data 1–3*). Therefore, it is possible that in response to auxin, triglycerides are mobilized by an unknown mechanism resulting in upregulation of fatty acid metabolism and attenuation of lipid breakdown. It is also possible that decreased interactions or feeding time with auxin-containing food promotes fasting, resulting in decreased TAG and lipid synthesis. Collectively, these results suggest that auxin exposure in adults significantly alters the transcriptome to adjust to decreased fatty acid metabolism and lipid stores.

Auxin is an essential hormone for plant growth and development (*Du et al., 2020*; *Wójcik et al., 2020*; *Gomes and Scortecci, 2021*), and has been widely used to manipulate gene expression in multiple organisms (*Zhang et al., 2015*; *Trost et al., 2016*; *Chen et al., 2018*; *Li et al., 2019*; *Shetty et al., 2019*; *Yesbolatova et al., 2020*; *Macdonald et al., 2022*). In *Arabidopsis*, auxin is a known regulator of fatty acid synthesis in plants (*He et al., 2020*) and has been shown to induce lipid synthesis to control vacuole trafficking (*Li et al., 2015*). Our results suggest that auxin has the opposite effect on lipid metabolism in adult *Drosophila* and decreases triglyceride levels. Given its widespread use, the role of auxin in potentially regulating lipid metabolism should be further characterized in additional

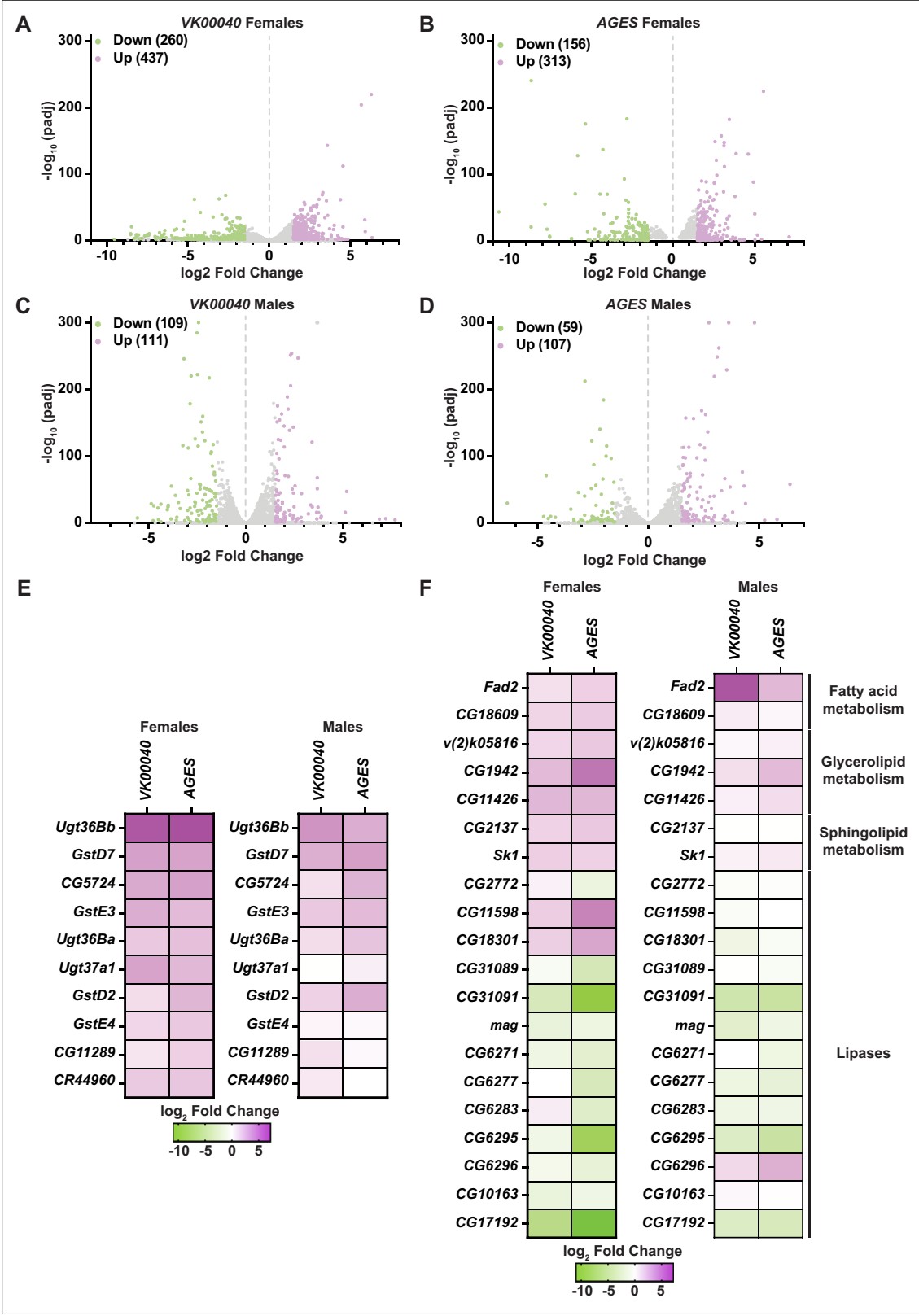

**Figure 3.** Lipid metabolism and detoxification transcripts are significantly altered in adult *Drosophila* in response to auxin exposure. Volcano plots of differentially expressed genes graphing the statistical significance [$-\log_{10}(p_{adj})$] against the magnitude of differential expression ($\log_2$ fold change) in females (**A**, **B**) or males (**C**, **D**) of the *VK00040* or *AGES* line. (**E**) Significantly upregulated genes involved in 'drug metabolism' in adult females or males of the *VK00040* or *AGES* lines exposed to 10 mM auxin and compared to the 0 mM control. (**F**) Significantly up- (purple) and downregulated (green)

*Figure 3 continued on next page*

*Figure 3 continued*

genes involved in fatty acid metabolism in adult females or males of the *VK00040* or *AGES* lines exposed to 10 mM auxin and compared to the 0 mM control.

The online version of this article includes the following source data and figure supplement(s) for figure 3:

**Source data 1.** RNA sequencing raw data from *VK00040* females.

**Source data 2.** RNA sequencing raw data from *AGES* females.

**Source data 3.** RNA sequencing raw data from *VK00040* males.

**Source data 4.** RNA sequencing raw data from *AGES* males.

**Figure supplement 1.** Transcripts of genes that regulate fatty acid metabolism and drug detoxification are altered in *Oregon-R* males and females exposed to auxin.

**Figure supplement 1—source data 1.** RNA sequencing raw data from *Oregon-R* females.

**Figure supplement 1—source data 2.** RNA sequencing raw data from *Oregon-R* males.

**Figure supplement 2.** Auxin exposure differentially regulates detoxification and fatty acid metabolism genes.

**Figure supplement 2—source data 1.** Quantitative reverse-transcriptase polymerase chain reaction (qRT-PCR) analysis from *Oregon-R*, *VK00040*, and *AGES* males and females.

model organisms using auxin to manipulate gene expression (at their respective recommended dosages), such as *C. elegans* (*Zhang et al., 2015*) and mice (*Macdonald et al., 2022*).

## Auxin exposure in adult females does not influence oogenesis

Although many tissues maintain and regulate lipid stores in *Drosophila* (including muscle and gut; reviewed in *Heier and Kühnlein, 2018*), the adult adipose tissue is a major lipid storage depot (reviewed in *Chatterjee and Perrimon, 2021*) and impacts peripheral tissue function such as adult *Drosophila* oogenesis (*Armstrong et al., 2014*; *Matsuoka et al., 2017*; *Armstrong and Drummond-Barbosa, 2018*; *Weaver and Drummond-Barbosa, 2018*; *Weaver and Drummond-Barbosa, 2019*). Thus, we sought to examine if the metabolic effects of auxin described above impact oogenesis.

Each ovary consists of 16–20 ovarioles composed of progressively older follicles that ultimately give rise to a mature egg chamber (*Figure 4A, B*; *Drummond-Barbosa, 2019*). Oogenesis is maintained by two to three germline stem cells (GSCs) that reside in the anterior germarium of each ovariole and can be identified based on their proximity to the stem cell niche, which is primarily composed of somatic cap cells. GSCs self-renew and give rise to early GSC progeny that differentiate to produce follicles that bud from the germarium and complete oogenesis.

To determine if auxin exposure influences fecundity or progeny survival, we performed egg laying and larval hatching analyses. Adult females (*Oregon-R*, *VK00040*, or *AGES*) were maintained with *AGES* males (due to their increased resistance to auxin) on molasses plates supplemented with wet inactive yeast containing either 0 mM auxin or 10 mM auxin for 15 days. Relative to 0 mM auxin control, exposure to 10 mM auxin did not significantly influence the number of eggs laid per female in any tested genotype (*Figure 4—figure supplement 1A, C, E*). We note that the number of eggs laid under 0 mM auxin conditions is relatively low; however, this is likely due to the decrease in egg production in females fed inactive yeast compared to active yeast paste (*Figure 4—figure supplement 2*). In addition, the relative hatching percentages of each genotype also were not affected by auxin exposure (*Figure 4—figure supplement 1B, D, F*). These results suggest that auxin exposure in adult females does not significantly influence the number of laid eggs or oocyte quality.

We next analyzed specific processes of oogenesis that are sensitive to metabolic changes including GSC maintenance, early germline cyst survival, and survival of vitellogenic follicles (*Drummond-Barbosa, 2019*). We detected GSCs by the morphology and location of the fusome relative to the GSC-niche (*Figure 4C*). GSC numbers in all genotypes exposed to 10 mM auxin were comparable to those in 0 mM auxin controls at each time point (*Figure 4D, F*; *Figure 4—figure supplement 3A, D*). Likewise, there was no change in the number of cap cells over time (*Figure 4—figure supplement 3E*). Furthermore, there were no effects of auxin exposure on the survival of early germline cysts as determined by ApopTag TUNEL labeling (*Drummond-Barbosa, 2019*; *Figure 4F–H*; *Figure 4—figure supplement 3B*) or survival of vitellogenic egg chambers (*Figure 4I–K*; *Figure 4—figure supplement 3C*). Based on these data, we conclude that the recommended auxin concentration for AGES in adult

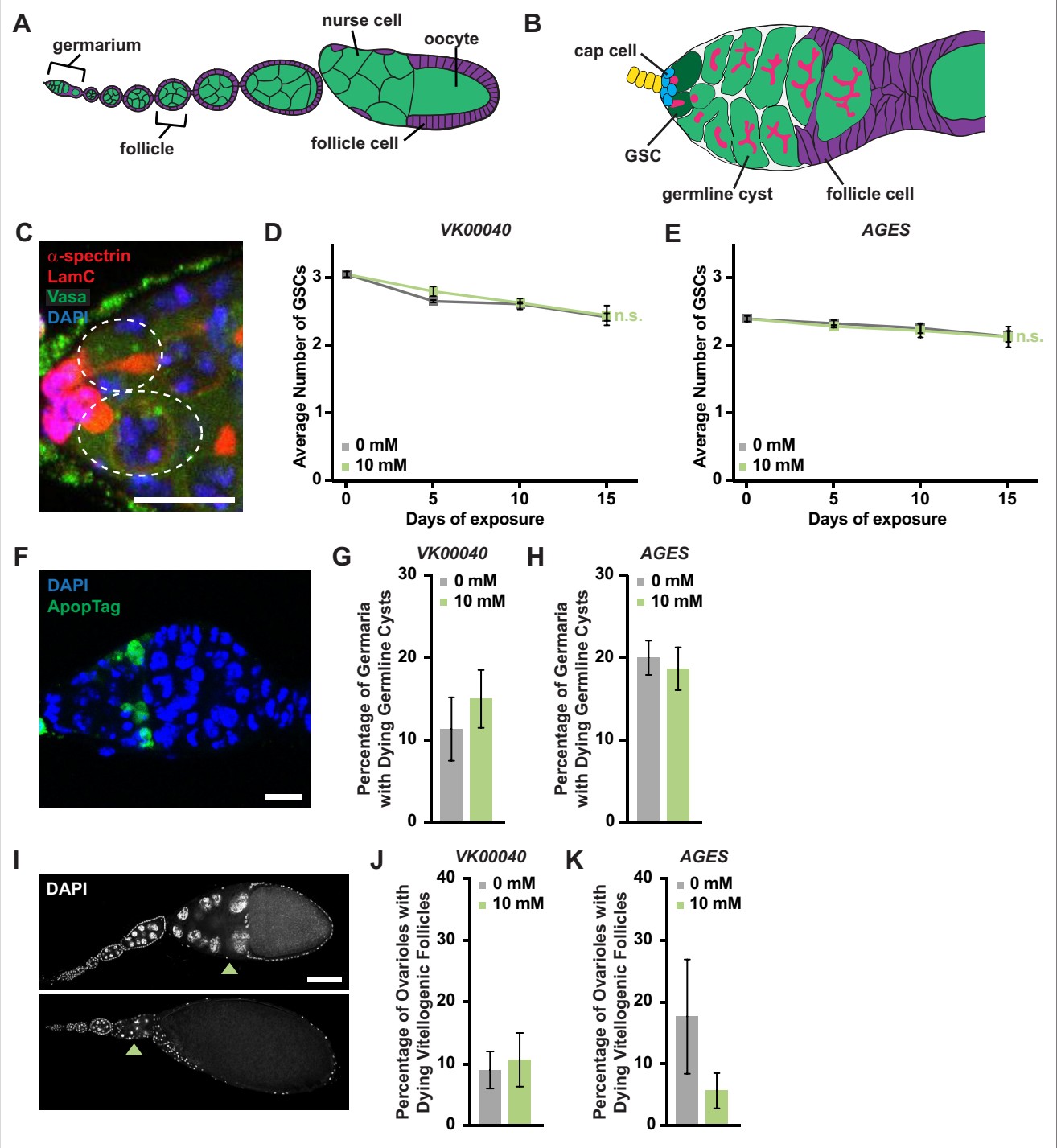

**Figure 4.** Auxin exposure does not significantly influence processes of oogenesis. (**A**) Cartoon schematic of the adult *Drosophila* ovariole showing the anterior germarium followed by developing egg chambers, which consist of 16 germ cells (15 nurse cells and 1 oocyte; green) that are surrounded by follicle cells (purple). (**B**) Schematic of the germarium, which contains two to three germline stem cells (GSCs, dark green) and somatic cells (gray and purple). Each GSC divides asymmetrically to self-renew and generate a cystoblast that divides to form a 16-cell cyst. Early germline cysts are surrounded by follicle cells (purple) to bud a new egg chamber. GSCs and their progeny are identified based on the position and morphology of the fusome (pink), a germline-specific organelle. (**C**) Germaria from adult females exposed to 10 mM auxin for 10 days. α-Spectrin (magenta), fusome; LamC (red), cap cell nuclear lamina; Vasa (green), germ cells; 4′,6-diamidino-2-phenylindole (DAPI; blue), nuclei. GSCs are outlined. Scale bar, 10 µm. Average number of GSCs per germarium over time in females exposed to 0 or 10 mM auxin in the *VK00040* (**D**) *AGES* (**E**) lines. At least 100 germaria were analyzed per time point and condition. Data shown as mean ± standard error of the mean (SEM). No statistically significant differences, two-way analysis of variance (ANOVA) with interaction. (**F**) Germaria from adult females exposed to 10 mM auxin. ApopTag (green), dying cells; DAPI (blue, nuclei). Average

*Figure 4 continued on next page*

*Figure 4 continued*

percentage of germaria containing ApopTag-positive germline cysts in adult females in the *VK00040* (**G**) or *AGES* (**H**) lines exposed to 0 or 10 mM auxin. Data shown as mean ± SEM; Student's *t*-test. 100 germaria were analyzed for each genotype and condition. (**I**) Ovarioles exposed to 10 mM auxin for 10 days showing a healthy ovariole (top) and an ovariole with a dying vitellogenic egg chamber (bottom). Arrowheads point to healthy or dying vitellogenic egg chambers. DAPI (white), nuclei. Scale bar, 100 μm. Average percentages of ovarioles containing dying vitellogenic egg chambers in females exposed to 0 or 10 mM auxin in the *VK00040* (**J**) or *AGES* (**K**) lines. Data shown as mean ± SEM, Student's *t*-test. 100 ovarioles were analyzed for each genotype and condition.

The online version of this article includes the following source data and figure supplement(s) for figure 4:

**Source data 1.** Germline stem cell (GSC) and cap cell analysis from *Oregon-R*, *VK00040*, and *AGES* females.

**Source data 2.** ApopTag analysis from *VK00040* females.

**Source data 3.** ApopTag analysis from *AGES* females.

**Source data 4.** Dying vitellogenic egg chamber analysis from *VK00040* females.

**Source data 5.** Dying vitellogenic egg chamber analysis from *AGES* females.

**Figure supplement 1.** Auxin exposure does not significantly alter egg laying or progeny survival.

**Figure supplement 1—source data 1.** Egg count analysis from *Oregon-R*, *VK00040*, and *AGES* females.

**Figure supplement 1—source data 2.** Hatching percentage analysis from *Oregon-R*, *VK00040*, and *AGES* females.

**Figure supplement 2.** Inactive yeast paste decreases egg laying in adult females.

**Figure supplement 2—source data 1.** Egg count analysis from *Oregon-R*, *VK00040*, and *AGES* females treated with active or inactive yeast paste.

**Figure supplement 3.** Processes of oogenesis are not significantly influenced by exposure to auxin.

**Figure supplement 3—source data 1.** ApopTag analysis from *Oregon-R* females.

**Figure supplement 3—source data 2.** Dying vitellogenic egg chamber analysis from *Oregon-R* females.

females does not adversely affect oogenesis, despite significant decreases in whole-body fatty acid metabolism and lipid composition.

Consistent with our results, it was recently shown that obese adult *Drosophila* females do not have reduced fecundity, but that fertility defects manifest only when combined with a high sugar diet (*Nunes and Drummond-Barbosa, 2023*). Collectively, these results suggest that lipid content alone (lean or obese) is not sufficient to regulate distinct processes of oogenesis in adult *Drosophila* females. Therefore, AGES may be suitable for the study of some processes in adult oogenesis.

## Conclusions and suggestions for future studies

The *Gal4/UAS* system has revolutionized the ability to perform tissue-specific manipulations in *Drosophila*. However, finding the ideal conditions to manipulate tissues temporally without causing significant alterations in physiology or impacting organism behavior presents a challenge. For example, using *Gal4/UAS* in conjunction with *Gal80ᵗˢ* for adult-specific manipulations has the adverse effect of decreasing adult female fecundity, making this system unsuitable for aging studies on oogenesis (*Gandara and Drummond-Barbosa, 2022*). Similarly, controlling transgene expression using RU486-inducible *GeneSwitch* drivers has not only been found to be less than ideal for feeding and aging studies (*Landis et al., 2015*; *Yamada et al., 2017*), but also presents a workplace hazard to pregnant researchers (*Avrech et al., 1991*). Likewise, our findings herein indicate that current methods requiring auxin for temporal transgene induction or protein degradation may work well to study many biological processes, but may not be the ideal system for the study of lipogenesis or other metabolic processes dependent on lipid content. Indeed, our data suggest that auxin-induced changes to metabolism persist even after its withdrawal from the diet, indicating that even short-term auxin treatments may have unwanted physiological effects.

Each modification to the *Gal4/UAS* system designed to provide temporal regulation of transgene expression has strengths and weaknesses and should be carefully vetted to ensure the most ideal experimental design is balanced with the caveats of the methodology. Regardless of the system used, researchers should determine whether their experimental manipulation alters Gal4 expression patterns compared to control conditions (i.e., thoroughly analyze Gal4 expression pattern across developmental time and tissues; *Weaver et al., 2020*). For example, the expression of transgenes using numerous neuronal Gal4 drivers with AGES is significantly weaker than Gal4-induced expression at 30°C (*Hawley et al., 2023*). Finally, researchers should ensure that non-specific transgene controls

used in the same conditions as the experimental (e.g., 0 versus 10 mM auxin) do not have phenotypes for physiological outputs of interest and ensure that changes in physiology that could confound result interpretations are accounted for.

# Materials and methods

**Key resources table**

| Reagent type (species) or resource | Designation | Source or reference | Identifiers | Additional information |
|---|---|---|---|---|
| Genetic reagent (*D. melanogaster*) | *Oregon-R* | Bloomington *Drosophila* Stock Center | BDSC:25211 | |
| Genetic reagent (*D. melanogaster*) | *w^1118* | Bloomington *Drosophila* Stock Center | BDSC:3605 | |
| Genetic reagent (*D. melanogaster*) | *w^1118; tubP-TIR1-T2A-Gal80.AID* | Bloomington *Drosophila* Stock Center | BDSC:92470 | |
| Genetic reagent (*D. melanogaster*) | *y^1 w^1118; VK00040/TM6B* | Bloomington *Drosophila* Stock Center | BDSC:9755 | |
| Genetic reagent (*D. melanogaster*) | *3.1Lsp2-Gal4^ts* | **Armstrong et al., 2014** | | |
| Genetic reagent (*D. melanogaster*) | *3.1Lsp2-Gal4^AGES* | This paper | | Fly line maintained in L. Weaver lab |
| Antibody | anti-alpha-spectrin (Mouse monoclonal) | Developmental Studies Hybridoma Bank | 3A9 (323 or M10-2) | IF (3 µg/ml) |
| Antibody | anti-LamC (Mouse monoclonal) | Developmental Studies Hybridoma Bank | LC28.26 | IF (0.8 µg/ml) |
| Antibody | anti-Vasa (Rat monoclonal) | Developmental Studies Hybridoma Bank | | IF (2.15 µg/ml) |
| Sequence-based reagent | *Rp49*_F1 | **Weaver and Drummond-Barbosa, 2018** | PCR primer | CAGTCGGATCGATATGCTAAGC |
| Sequence-based reagent | *Rp49*_R1 | **Weaver and Drummond-Barbosa, 2018** | PCR primer | AATCTCCTTGCGCTTCTTGG |
| Sequence-based reagent | *Act5C*_F1 | This paper | PCR primer | AGGCCAACCGTGAGAAGATG |
| Sequence-based reagent | *Act5C*_R1 | This paper | PCR primer | ACATACATGGCGGGTGTGTT |
| Sequence-based reagent | *GstD7*_F1 | This paper | PCR primer | TATCCACCGTAGAATGGTTTTCG |
| Sequence-based reagent | *GstD7*_R1 | This paper | PCR primer | CCCATCCAGGAGTTACTTTTGG |
| Sequence-based reagent | *GstE3*_F1 | This paper | PCR primer | ATGGGAAAACTTACGCTTTACGG |
| Sequence-based reagent | *GstE3*_R1 | This paper | PCR primer | ACGATCTTGTAGTCGAAGTCCA |
| Sequence-based reagent | *Fad2*_F1 | This paper | PCR primer | ACCGGAGTGCTTTACGAATCC |

*Continued on next page*

*Continued*

| Reagent type (species) or resource | Designation | Source or reference | Identifiers | Additional information |
|---|---|---|---|---|
| Sequence-based reagent | *Fad2*_R1 | This paper | PCR primer | GTGGTCTTCCGATTGCTTAGC |
| Sequence-based reagent | *mag*_F1 | This paper | PCR primer | GCATCCCGTACTCCCACAAG |
| Sequence-based reagent | *mag*_R1 | This paper | PCR primer | AGTTACTGAACAATCCGTGCTG |
| Commercial assay or kit | Stanbio Triglyceride Liquicolor kit | Thermo Fisher Scientific | Cat#SB-2100-430 | |
| Commercial assay or kit | Invitrogen RNAqueous-4PCR DNA-free RNA isolation for RT-PCR kit | Thermo Fisher Scientific | Cat#AM1914 | |
| Commercial assay or kit | NEBNext Ultra II RNA Library Prep Kit for Illumina | New England Biolabs | Cat# E7770L | |
| Commercial assay or kit | ApopTag Indirect In Situ Apoptosis Detection Kit | Thermo Fisher Scientific | Cat#S7110 | |
| Chemical compound, drug | 1-Naphthaleneacetic acid | Santa Cruz Biotechnology | Cat#15165-79-4 | Dissolved in 1 mM NaOH |
| Software, algorithm | Benchmark Dose Software (BMDS) | U.S. Environmental Protection Agency | https://www.epa.gov/bmds | |
| Software, algorithm | FLIC source code | *Ro et al., 2014*; *Ro et al., 2020* | https://github.com/PletcherLab/FLIC_R_Code | |
| Software, algorithm | MetaboAnalyst 5.0 | *Pang et al., 2021* | https://www.metaboanalyst.ca/ | |
| Other | Superscript II Reverse Transcriptase | Thermo Fisher Scientific | Cat# 18064014 | Reverse transcriptase used for cDNA synthesis |
| Other | PowerUp SYBR Green Master Mix | Thermo Fisher Scientific | Cat# A25742 | Master mix used for qPCR reactions |
| Other | VECTASHIELD (Antifade Mounting Medium with DAPI) | Thermo Fisher Scientific | Cat# NC9524612 | Anti-fade mounting media with DAPI included |

### *Drosophila* strains and culture

*Drosophila* stocks were maintained on Bloomington *Drosophila* Stock Center (BDSC) Cornmeal Food that consists of 15.9 g/l inactive yeast, 9.2 g/l soy flour, 67 g/l yellow cornmeal, 5.3 g/l agar, 70.6 g/l light corn syrup, 0.059 M propionic acid at 22–25°C. Medium was supplemented with inactive wet yeast paste for all experiments, unless otherwise noted. For a subset of experiments, where indicated, flies were reared on yeast–sugar–cornmeal food (*Lewis, 1960*) that consists of 20.5 g/l white sugar, 70.9 g/l D-glucose, 48.5 g/l cornmeal, 30.3 g/l yeast, 0.5 g/l $CaCl_2$-$2H_2O$, 0.5 g/l $MgSO_4$-$7H_2O$, 4.6 g/l agar, 4.9 ml/l propionic acid, and 0.49 ml/l phosphoric acid. For experiments using this diet, the food was not supplemented with inactive wet yeast paste. The previously described $w^{1118}$; *tubP-TIR1-T2A-Gal80.AID* '*AGES*' (*McClure et al., 2022*) line was used and obtained from the Bloomington *Drosophila* Stock Center (BDSC 92470; https://bdsc.indiana.edu). *Oregon-R* (BDSC 25211) and $y^1$ $w^{1118}$; *VK00040/TM6B* (BDSC 9755) lines were used as controls. Flies from the $w^{1118}$ strain (BDSC 3605) were used to monitor larval development and adult triglyceride levels. Balancer chromosomes and other genetic elements are described in Flybase (https://www.flybase.org/).

For all ovarian experiments, 0- to 2-day-old females were mated with *AGES* males and incubated at 25°C for up to 15 days at ≥70% humidity on medium containing either 1 mM NaOH 0 mM 1-naphthaleneacetic acid (referred to as 'auxin' throughout the manuscript; Santa Cruz Biotechnology) or 10 mM auxin dissolved in 1 mM NaOH the recommended concentration of auxin (*McClure et al., 2022*). Medium was supplemented with inactive wet yeast containing either 1 mM NaOH or 10 mM auxin daily, except where noted.

## Larval development time

$w^{1118}$ females laid eggs on grape plates supplemented with yeast paste during a 3-hr collection period. Newly hatched larvae were transferred to the yeast–sugar–cornmeal diet at a density of 50 larvae per 10 ml food. For the auxin-containing medium, auxin was added to the recommended concentration of 5 mM to cooled fly food immediately prior to filling vials (*McClure et al., 2022*). Percent pupation was calculated by comparing the number of pupae at each 12-hr interval to the total pupae in the vial.

## Auxin exposure dose–response curves

Dose–response curves were performed as previously described (*Holsopple et al., 2023*). Briefly, 0- to 2-day-old flies were transferred to bottles containing fresh BDSC food and aged for 2 days. Flies were sorted by sex in groups of 20 per vial and aged for an additional 48 hr at 25°C to allow recovery from $CO_2$ anesthesia. Flies were then transferred to starvation vials containing sterile milli-Q water for 16 hr at 25°C. Following starvation, flies were transferred to exposure vials containing liquid food [4% sucrose (m/v), 1.5% yeast extract (m/v), 1 mM NaOH] and 0, 2, 4, 6, 10, 15, 20, 25, 90, 125, or 350 mM auxin. The number of dead female or male flies per vial was counted in six replicates per concentration and the percentage of dead flies was subjected to mathematical modeling through the Benchmark Dose Software (BMDS) published by the U.S. Environmental Protection Agency (EPA; https://www.epa.gov/bmds).

## FLIC assay

The FLIC system was used to determine differences in feeding behavior in male and female flies exposed to food with or without auxin as previously described (*Ro et al., 2014*). FLIC *Drosophila* Feeding Monitors (DFMs, Sable Systems International, models DFMV2 and DFMV3) were used in the single choice configuration and each chamber was loaded with 0 mM auxin liquid food solution [4% sucrose (m/v), 1.5% yeast extract (m/v), 1 mM NaOH] or the recommended dose of 10 mM auxin liquid food [4% sucrose (m/v), 1.5% yeast extract (m/v), 10 mM auxin]. Four- to six-day-old flies were briefly anesthetized and aspirated into the DFM chambers. Feeding behavior was measured for 24 hr. Each FLIC experiment contains pooled data from at least 30 flies for each genotype, sex, and auxin concentration condition. FLIC data were analyzed using previously described custom R code (*Ro et al., 2014*; *Ro et al., 2020*). Default thresholds were used for analysis except for the following: minimum feeding threshold = 10, tasting threshold = (0,10). Animals that did not participate (i.e., returned zero values), whose DFM returned an unstable baseline signal, or who produced extreme outliers (i.e., exceeding twice the mean of the population) were excluded from analysis. Data were subjected to a Mann–Whitney *U*-test.

## Ultra high-pressure liquid chromatography–mass spectrometry-based metabolomics and analysis

Adult male and female flies were exposed to 0 mM or the recommended dose of 10 mM auxin as described above for the dose–response curve assays and were flash frozen in liquid nitrogen. Analyses were performed at the University of Colorado Anschutz Medical Campus, as previously described (*Nemkov et al., 2019*; *Nemkov et al., 2022*) with minor modifications. Briefly, the analytical platform employs a Vanquish UHPLC system (Thermo Fisher Scientific) coupled online to a Q Exactive mass spectrometer (Thermo Fisher Scientific). The (semi)polar extracts were resolved over a Kinetex C18 column, 2.1 × 30 mm, 1.7 µm particle size (Phenomenex) using a high-throughput 1-min gradient. Solvents were supplemented with 0.1% formic acid for positive mode runs and 10 mM ammonium acetate + 0.1% ammonium hydroxide for negative mode runs. The Q Exactive mass spectrometer (Thermo Fisher Scientific) was operated independently in positive or negative ion mode, scanning in Full MS mode (2 µscans) from 60 to 900 *m/z* at 70,000 resolution, with 4 kV spray voltage, 45 sheath gas, 15 auxiliary gas. Calibration was performed prior to analysis using the Pierce Positive and Negative Ion Calibration Solutions (Thermo Fisher Scientific). Metabolomics raw data were processed using El-Maven (*Agrawal et al., 2019*) and analyzed using MetaboAnalyst 5.0 (*Pang et al., 2021*), with the data first preprocessed using log normalization and Pareto scaling.

## Triglyceride assays

Zero- to two-day-old adult males and females were collected and maintained on BDSC food for 2 days. Two- to four-day-old adults were starved for 16 hr and then exposed to 0 mM or the recommended dose of 10 mM auxin liquid food for 48 hr as described above for the dose–response assays. Whole bodies from five animals of each sex and genotype were washed in phosphate-buffered saline (PBS), pH 7.0 and flash frozen in liquid nitrogen. Samples were homogenized in 100 μl cold PBS + 0.05% Tween 20 (PBST) and heat treated for 10 min at 90°C. The resulting homogenate was assayed for triglyceride (TAG) and soluble protein levels as previously described (*Tennessen et al., 2014*). TAG amounts were normalized to protein amounts and expressed as μg/ml TAG per μg/ml protein. Data were subjected to a paired Student's *t*-test.

For TAG assays on *w^{1118}* flies, newly hatched larvae were transferred to yeast–sugar–cornmeal food and reared at a density of 50 larvae per 10 ml food at 25°C. Male and female pupae were separated as late pupae according to sex combs. Two experimental designs were used. For one design, virgin male and female flies were kept at a density of 20 flies per 10 ml food (+/− recommended dose of 10 mM auxin) from eclosion until 5 days of age. Five-day-old male and female flies were collected, snap frozen at −80°C, weighed, and finally subjected to a TAG assay. In the second experimental design, newly eclosed flies were subjected to one of three protocols: (1) maintained continuously on yeast–sugar–cornmeal food with no auxin, where flies were collected, frozen, weighed, and subjected to a TAG assay at 5 and 10 days of age; (2) maintained on yeast–sugar–cornmeal food with 10 mM auxin for 5 days and shifted to food with no auxin for a further 5 days, where flies were collected, frozen, weighed, and subjected to a TAG assay at 5 and 10 days of age; and (3) maintained continuously on yeast–sugar–cornmeal food with the recommended 10 mM dose of auxin, with flies collected, frozen, weighed, and subjected to a TAG assay at 5 and 10 days of age. Flies were flipped every 2 days in both experimental designs. One biological replicate of three or five flies was homogenized in 150 or 350 μl of 0.1% Tween in 1× PBS using 50 μl of glass beads agitated at 8 m/s for 5 s. TAG assay was performed using the Stanbio Triglyceride Liquicolor kit (Stanbio) according to the manufacturer's protocol. TAG is expressed as percent body fat as previously described (*Wat et al., 2020*). Each experiment includes four biological replicates, and each experiment was repeated twice for a total of eight biological replicates per sex; data were analyzed using either a Student's *t*-test, one-way ANOVA, or two-way ANOVA, as indicated.

## RNA isolation, RNA sequencing, and data analysis

Twenty whole adult animals of each genotype and sex were exposed to 0 mM or the recommended dose of 10 mM auxin, as described above for the dose–response curves, and flash frozen in liquid nitrogen. Tissue was lysed in 500 μl lysis buffer from the RNAqueous-4PCR DNA-free RNA isolation for RT-PCR kit (Invitrogen). RNA was extracted from all samples following the manufacturer's instructions. Three independent experiments were performed for RNA sequencing.

cDNA library construction, Illumina sequencing, and differential expression analysis were performed by Novogene Bioinformatics Technology Co, Ltd (Beijing, China). The cDNA libraries were prepared using the NEBNext Ultra II RNA Library Prep Kit for Illumina (New England Biolabs) according to the manufacturer's instructions. The cDNA library for each sample was quality assessed using an Agilent Bioanalyzer 2100, and library preparations were sequenced on a NovaSeq6000 platform with PE150 read lengths.

Reads obtained from sequencing were aligned to the *D. melanogaster* reference genome using the TopHat read alignment tool (*Trapnell et al., 2009*) for each of the sequencing datasets. The reference sequences were downloaded from the Ensembl project website (https://useast.ensembl.org). TopHat alignments were used to generate read counts for each gene using HTSeq (*Anders et al., 2015*), which were subsequently used to generate the differential expression results using the DESeq2 R package (*Anders et al., 2015*). RNA sequencing produced an average of 40,448,108 reads across the 36 sequencing libraries, ranging from 37,632,234 to 58,099,256 reads per sample (representing an average of 96.2% mapped to the *Drosophila* genome). Enriched genes with a corrected p value less than 0.05 were considered significant.

## cDNA synthesis and quantitative reverse-transcriptase polymerase chain reaction

cDNA was synthesized from 500 ng of total RNA described above for each sample using Superscript II Reverse Transcriptase (Thermo Fisher Scientific) according to the manufacturer's instructions.

PowerUp SYBR Green Master Mix (Thermo Fisher Scientific) was used for RT-qPCR. The reactions for three independent biological replicates were performed in triplicate using LightCycler 96 (Roche). Amplification fluorescence threshold was determined by LightCycler 96 software, and ddCT were calculated using Microsoft Excel. Fold change of transcript levels was calculated in Excel as described (*Taylor et al., 2019*). The primers used for all PCR reactions are listed in the "Key Resources Table." *Rp49* and *Act5C* transcript levels were used as references.

### Egg laying and hatching assays

Egg production was measured as previously described (*Weaver and Drummond-Barbosa, 2019*) by maintaining five experimental females mated with five *AGES* males in perforated plastic bottles capped with molasses/agar plates smeared with 0 mM auxin or the recommended concentration of 10 mM auxin inactive yeast paste. Molasses/agar plates were changed twice daily. The number of eggs laid per day was counted in five replicates per genotype and results were subjected to a paired Student's *t*-test.

Egg hatching was measured by transferring up to 30 eggs from molasses/agar plates to fresh molasses/agar plates containing 0 mM auxin inactive yeast paste in the center every 2 days and the number of eggs that has hatched were counted 24 hr after the transfer. The number of eggs hatched per day was counted in five replicates per genotype and results were subjected to a paired Student's *t*-test.

### Adult female ovary immunostaining and fluorescence microscopy

Ovaries and carcasses were dissected in Grace's Insect Medium (Gibco), fixed, and washed as previously described (*Weaver and Drummond-Barbosa, 2019*). Samples were blocked for at least 3 hr in 5% normal goat serum (Jackson ImmunoResearch) and 5% bovine serum albumin (Sigma) in PBS [10 mM $NaH_2PO_4$/$NaHPO_4$, 175 mM NaCl (pH 7.4)] containing 0.1% Triton X-100 (PBST). Samples were incubated overnight at 4°C in primary antibodies diluted in blocking solution as follows: mouse monoclonal anti-alpha-spectrin (Developmental Studies Hybridoma Bank; DSHB, 3 µg/ml), mouse monoclonal anti-LamC (DSHB, 0.8 µg/ml), and rat monoclonal anti-Vasa (DSHB, 2.15 µg/ml). Samples were washed in PBST and incubated at room temperature for 2 hr with 1:200 Alexa Fluor 488- or 568-conjugated goat-species-specific secondary antibodies (Molecular Probes) in blocking solution. Samples were then washed three times for 15 min and mounted in Vectashield containing 1.5 µg/ml 4',6-diamidino-2-phenylindole (DAPI; Vector Laboratories) and imaged using a Leica SP8 confocal.

Cap cells and GSCs were identified as described (*Weaver and Drummond-Barbosa, 2019*), and two-way ANOVA with interaction (GraphPad Prism) was used to calculate the statistical significance of any differences among genotypes in the rate of cap cell or GSC loss from at least three independent experiments, as described (*Armstrong et al., 2014*). Progression through vitellogenesis was assessed using DAPI staining, as described (*Weaver and Drummond-Barbosa, 2019*). Three independent experiments were performed and subjected to a Student's *t*-test for statistical analysis.

### ApopTag assays

To detect dying germline cysts, the ApopTag Indirect In Situ Apoptosis Detection Kit (Millipore Sigma) was used according to the manufacturer's instructions as previously described (*Weaver and Drummond-Barbosa, 2019*). Briefly, fixed and teased ovaries were rinsed in equilibration buffer twice for 5 min each at room temperature. Samples were incubated in 100 µl TdT solution at 37°C for 1 hr with mixing at 15-min intervals. Ovaries were washed three times in 1× PBS followed by incubation in anti-digoxigenin conjugate for 30 min at room temperature protected from light. Samples were washed four times in 1× PBS and processed for immunofluorescence as described above.

## Acknowledgements

We thank the Bloomington Stock Center (National Institutes of Health P40OD018537) for *Drosophila* stocks. We are thankful to Flybase, an essential *Drosophila* research resource (NIH 5U41HG000739). The authors would like to thank the Indiana University Light Microscopy Imaging Center (LMIC) for access to microscopy facilities. We are grateful for Kristina J Weaver and Scott D Pletcher for assistance with the FLIC code. We are also grateful to Elizabeth T Ables, Brian R Calvi, and Deepika Vasudevan for critical reading of the manuscript. This work was supported by the Canadian Institutes

of Health Research grant PJT-153072 (EJR and PB), the National Institutes of Health (NIH) grants R35 GM119557 (JMT), R00 GM127605 (LNW), and R35 GM150517 (LNW).

## Additional information

### Funding

| Funder | Grant reference number | Author |
|---|---|---|
| National Institutes of Health | R35 GM150517 | Lesley N Weaver |
| National Institutes of Health | R00 GM127605 | Lesley N Weaver |
| National Institutes of Health | R35 GM119557 | Jason M Tennessen |
| Canadian Institutes of Health Research | PJT-153072 | Elizabeth Rideout |

The funders had no role in study design, data collection, and interpretation, or the decision to submit the work for publication.

### Author contributions

Sophie A Fleck, Performed experiments and contributed to data analysis; Puja Biswas, Performed experiments, contributed to data analysis, provided writing for the original draft, and provided edits; Emily D DeWitt, Performed experiments; Rebecca L Knuteson, Performed experiments and provided comments on the original draft; Robert C Eisman, Performed experiments and provided comments on the original draft; Travis Nemkov, Performed metabolomics analysis and provided comments on the original draft; Angelo D'Alessandro, Supervised and performed metabolomics analysis. Also provided comments on the original draft; Jason M Tennessen, Conceptualized the project, acquired funding for the project, supervised the metabolomics and RNA sequencing experiments, and provided comments on all drafts; Elizabeth Rideout, Conceptualized the project, acquired funding for the project, provided formal data analysis, supervised multiple aspects of the project, wrote sections of the original draft, and provided edits; Lesley N Weaver, Conceptualized the project, acquired funding for the project, performed experiments, acquired data, provided formal data analysis, supervised multiple aspects of the project, wrote the original draft, and provided edits

### Author ORCIDs

Sophie A Fleck (ID) https://orcid.org/0000-0002-0126-1160
Puja Biswas (ID) https://orcid.org/0000-0001-6808-6662
Travis Nemkov (ID) https://orcid.org/0000-0001-8566-7119
Jason M Tennessen (ID) https://orcid.org/0000-0002-3527-5683
Elizabeth Rideout (ID) https://orcid.org/0000-0003-0012-2828
Lesley N Weaver (ID) https://orcid.org/0000-0002-3120-8301

Reviewer #1 (Public Review): https://doi.org/10.7554/eLife.91953.3.sa1
Reviewer #2 (Public Review): https://doi.org/10.7554/eLife.91953.3.sa2
Reviewer #3 (Public Review): https://doi.org/10.7554/eLife.91953.3.sa3
Author Response https://doi.org/10.7554/eLife.91953.3.sa4

## Additional files

### Supplementary files

• MDAR checklist

## Data availability

*Drosophila* strains can be purchased from the Bloomington *Drosophila* Stock Center. The data and analyses in this paper are described in the main figures. The raw RNA sequencing data and processed data files are available through the NCBI GEO accession number GSE237283. Raw data are also provided as supplemental figures and source data.

The following dataset was generated:

| Author(s) | Year | Dataset title | Dataset URL | Database and Identifier |
|---|---|---|---|---|
| Weaver LN | 2023 | The effect of Auxin exposure on the adult *Drosophila* transcriptome | https://www.ncbi.nlm.nih.gov/geo/query/acc.cgi?acc=GSE237283 | NCBI Gene Expression Omnibus, GSE237283 |

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
