## [Editor Report · eLife assessment]

This **valuable** study shows that auxin exposure perturbs feeding behavior, survival rates, lipid metabolism, and gene expression patterns in adult *Drosophila* flies. The results are **solid** with proper methods and data analyses, and the evidence broadly supports the conclusions with only minor weaknesses. This work is relevant for fly geneticists who are interested in using the auxin-inducible gene expression system for inducing target protein degradation acutely.

---

## [Referee Report · Reviewer #1 (Public Review)]

In recent years, Auxin treatment is frequently used for inducing targeted protein degradation in *Drosophila* and various other organisms. This approach provides the way to acutely alter the levels of specific proteins. In this manuscript, the authors carefully examine the impact of Auxin treatment and provide strong evidence that Auxin treatment elicits alterations in feeding activity, survival rates, lipid metabolism, and gene expression patterns. Researchers need to be aware of these effects to design experiment/controls and interpret their data.

Strengths:

Regarding widespread usage of Auxin mediated gene manipulation method, it is important to address whether the application of Auxin itself causes any physiological changes. Authors provide evidence of several Auxin effects on lipid metabolism, feeding behavior and gene expression changes. Experiments are suitably designed with appropriate sample size, data analysis methods.

Weaknesses:

Data shown here are limited for certain method of treatment. No time course, dose dependency information is provided, and cell-type-specific responses are unknown. Therefore, this work basically provides the cautionary note for the field for researchers who use this method suggesting the importance that they should thoroughly check the gene expression pattern for their specific tissue of interest under their normal standard or altered food conditions.

---

## [Referee Report · Reviewer #2 (Public Review)]

In this study, Fleck and colleagues investigate the effects of auxin exposure on *Drosophila melanogaster* adults, focusing their analysis on feeding behavior, fatty acid metabolism, and oogenesis. The motivation for the study is that auxin-inducible transcription systems are now being used by *Drosophila* researchers to drive transcription using the Gal4-UAS system as a complement to Gal80ts versions of the system. I found the study to be carefully done. This study will be of interest for researchers using the *Drosophila* system, especially those focusing on fatty acid metabolism or physiology. The authors have adequately addressed all the minor points I raised in my review of the first submission.

---

## [Referee Report · Reviewer #3 (Public Review)]

This work by Fleck et al. and colleagues documented the auxin feeding-induced effects in adult flies, since auxin could be used in temporally control gene expression using a modified Gal4/Gal80 system. Overall, the experiments were well designed and carefully executed. The results were quantified with appropriate statistical analyses. The paper was also well written and the results were presented logically. Their findings demonstrate that auxin-fed flies have significantly lower triglyceride levels than the control flies using Ultra High-pressure Liquid Chromatography-Mass Spectrometry (UHPLC-MS)-based metabolomics assays. Further transcriptome analyses using the whole flies show changes of genes involved in fatty acid metabolism. However, female oogenesis and fecundity do not seem to be affected, at least using the current assays. These results indicate that auxin may not be used in experiments involving lipid-related metabolism, but could be appropriate to be applied for other biological processes. Researchers need to be careful when applying this strategy in their own experimental design and should perform proper controls.

---

## [Author Response]

The following is the authors’ response to the original reviews.

**REVIEWER 1:**

Reviewer 1 stated: “The authors have provided strong evidence that high levels of auxin exposure perturb feeding behavior, survival rates, lipid metabolism, and gene expression patterns, providing a cautionary note for the field in using this technology. They also concluded that “overall, the experiments were suitably designed with appropriate sample size and data analysis methods.”

Reviewer 1 provided the following recommendations for improvement, which are addressed below:

Point 1: “Although authors showed that auxin causes gene expression changes including the possible alteration of Gal4 expression levels, no cell-type-specific data is provided. It would be informative to the *Drosophila* field if the authors could examine major Gal4 drivers in their expression levels, such as the ones used in studying metabolism and oogenesis.”

We agree with the reviewer that cell-type specific Gal4 expression should be thoroughly analyzed by scientists in the community wishing to use the current auxin-inducible gene expression system (AGES) in their studies; however, those analyses are beyond the scope of our manuscript. There are many tissues and cell types that are used to study metabolism and oogenesis (e.g., muscle, adipocytes, oenocytes, multiple cell types in the gut, multiple cell types in the ovary), and Gal4 expression patterns could be different depending on age, sex, and diet. It is therefore impossible for us to pinpoint one or two key tissues important for regulating lipid levels and would be a significant investment of time. We believe that each researcher should thoroughly check the Gal4 expression pattern for their specific tissue of interest under their normal standard or altered food conditions. As this reviewer pointed out, our current study provides a cautionary note for the field in using this technology. Nevertheless, we have provided a reference to a recent micropub (Hawley et al; PMID: 37396791) which describes neuronal Gal4 expression patterns comparing the AGES and temporal and regional gene expression targeting (TARGET) systems and updated the text in lines 539-544 of the revised manuscript.

Point 2: “Although the authors briefly mentioned aging research, feeding behavior, and lipid metabolism, RNA-seq data are provided only for short-term treatment (2 days). The ovary phenotype was examined with long-term treatment (15 days). It would be informative if the authors could also show other long-term treatment data.”

We respectfully point out to the reviewer that a 5-day auxin feeding assay was provided in Figure S4H, which reproduces the data provided for the 2-day auxin treatment. In addition, the original AGES paper (McClure et al, PMID: 35363137) provided adult survival data that extended to 80 days. In our updated manuscript, we have provided data for a 10-day auxin treatment that also addresses Point #4 below regarding whether the decrease in lipid levels upon auxin feeding is reversible.

Point 3: “The auxin used in this work is a more water-soluble version and at a high concentration (10 mM). In the *C. elegans* system, researchers are using a much lower concentration of auxin typically at 1 mM. Therefore, the discussion of their results in terms of potential impacts on other experimental systems should be done carefully. It would be helpful to know what impacts might be observed at a lower concentration of auxin. The recommendation would be that the authors add the 1 mM auxin data point to key elements of their analysis.”

The concentration of 10 mM auxin used in our study is the recommended dose to use in *Drosophila* (see McClure et al) and has been used in at least one additional study (Hawley et al). We also would like to point out that other systems (e.g., *C. elegans* and mice) have many differences in physiology and therefore the concentration of auxin used to elicit a response are likely to be different (e.g., 71.4 mM final concentration is the recommended concentration used in mice; Macdonald et al; PMID: 35736539). We have merely suggested that researchers using auxin for protein degradation should carefully check whether lipid levels (or other physiological processes of interest) are altered upon auxin feeding (or soaking) alone compared to a 0 mM auxin control. The text in lines 467-470 has been altered to reflect this. In addition, the specific recommended dose for *Drosophila* is highlighted and referenced in multiple places (i.e., methods and results and discussion) throughout the updated text.

Point 4: “Another related question is whether these detected changes are reversible or not after exposure to auxin at different concentrations. This would be informative for researchers to better design their temporally controlled experiments.”

We thank the reviewer for this suggestion and have provided the data in Figure S4I. Briefly, we found that after a 5-day treatment of auxin, removal of auxin for an additional 5 days does not recover lipid levels to those of control animals never exposed to auxin.

Point 5: “It would also be helpful to know whether spermatogenesis is affected or not.”

Although this would be an interesting developmental process to determine if affected by auxin exposure, we believe that these analyses are beyond the scope of the current manuscript.

Point 6: “A few other points include changing the nomenclature and validating some of the key genes shown in Figure 3 using quantitative RT-PCR experiments with the tissues where the affected genes are known to be expressed and functional.”

We thank the reviewer for this suggestion. We have provided qRT-PCR analysis using whole body samples and this data is now provided in the new Figure S8. We used whole-body samples for the qRT-PCR analysis because it would be impossible to pinpoint the specific tissue the differentially regulated genes are required for eliciting the response to auxin exposure. For example, according to Flybase (flybase.org) GstE3 transcripts are moderately to highly expressed in 15 of the 23 cell types annotated by the Fly Cell Atlas project (Li et al; PMID: 35239393).

**REVIEWER 2:**

Reviewer 2 stated: “The authors provide evidence of several Auxin effects. Experiments are suitably designed with appropriate sample size and data analysis methods.”

This reviewer expressed the following concerns, which are addressed below:

Point 1: “The provided information is limited and not very helpful for many applications. For example, although authors briefly mentioned aging research, feeding behavior, and lipid data, RNA seq data are provided only for short-term (48 hours) treatment. Especially, since ovary phenotype was examined with long-term treatment (15 days), authors should also show other data for long-term treatment as well.”

Please see our response to Point #2 of Reviewer 1 regarding long-term treatment experiments. Furthermore, although the ending timepoint for the ovarian analyses is 15 days, we also provide analysis at shorter time points (e.g., daily analysis for egg counts, 5 and 10 day timepoints for fixed sample analyses).

Point 2: “Although the authors show that Auxin causes a change in gene expression patterns and suggests the possible alteration of Gal4 expression levels, no cell-type-specific data is provided. It would be informative if the authors could examine the expression level of major Gal4 drivers. Authors should discuss how severe these changes are by comparing them with other treatments or conditions, such as starvation or mutant data (ideally, comparing with reported data or their own data if any?).”

Please see our response to Point #1 from Reviewer 1.

**REVIEWER 3:**

Reviewer 3 stated that they “found the study to be carefully done” and “this study will be of interest to researchers using the *Drosophila* system, especially those focusing on fatty acid metabolism or physiology.”

Reviewer 3 also had the following minor points, which are addressed below:

Point 1: “Auxin, actually 1-naphthaleneaceid acid here, which is a more water-soluble version of auxin (indole-3-acetic acid) is used at what I consider to be a high concentration-10 mM. The problem I have is that the authors are discussing their results in terms of potential impacts on other experimental systems. At least for *C. elegans*, I think this is not a reasonable extension of the current dataset. In the *C. elegans* system, researchers are using 1 mM auxin. The authors note that their RNA-seq results suggest a xenobiotic response. Could this apparent xenobiotic response be due to a metabolic byproduct following auxin administration at high concentrations? Figure S1A shows that there is quite a robust transcriptional response at 1 mM auxin. It would be helpful to know what impacts might be observed at this lower concentration in which the transcriptional induction could be used in the context of biologically meaningful experiments. The recommendation would be that the authors add the 1 mM auxin data point to key elements of their analysis.”

Regarding the comparisons to other model organisms, we refer to our response to Point #3 from Reviewer 1. We also point out that although there is a robust response to 1 mM auxin using the 3.1Lsp2-Gal4 driver, 1 mM is not sufficient for a robust response using additional driver lines in *Drosophila* (see Hawley et al). It is possible that the xenobiotic response is due to using the recommended dose of auxin (McClure et al).

However, given the fact that researchers are currently using the 10 mM dose for experiments in *Drosophila*, we believe that the 10 mM transcription dataset is the most relevant. Nevertheless, we do agree that researchers who choose to use lower concentrations of auxin in the future should carefully look at whether any transcriptional induction alters physiological processes of interest.

Point 2: “This reviewer was confused by the genetic nomenclature the authors use. The authors have chosen to use the designation 3.1Lsp2-Gal4 (3.1Lsp2-Gal4AID). I think this is potentially confusing because a reader might think that it is the Gal4 transcription factor that is the direct target of auxin- and TIR1-mediated protein degradation, as I initially did. Rather, it is the Gal80 repressor protein that is the direct target. The authors might consider a nomenclature that is more reflective of how this system works. It would also be helpful if the full genotypes of strains were included in each figure legend.”

We apologize for the nomenclature confusion in our original submission. We have changed our “AID” nomenclature throughout the manuscript to “AGES,” which is the nomenclature used in McClure et al. We respectfully note that the traditional nomenclature for using the temperature-sensitive Gal80 system is Gal80ts or adding the “ts” superscript to the Gal4 line used (e.g., 3.1Lsp2ts).

Point 3: “The RNA-seq dataset does not appear to be validated by RT-PCR experiments. The authors should consider validating some of the key genes shown in Figure 3 using quantitative RT-PCR experiments, potentially adding a 1 mM auxin data point.”

Please see our response to Point #6 to Reviewer 1.

**REVIEWER 4:**

Reviewer 4 stated: “Overall, the experiments were well-designed and carefully executed. The results were quantified with appropriate statistical analyses. The paper was also well-written and the results were presented logically.”

**RECOMMENDATIONS FOR THE AUTHORS:**

We have further addressed reviewer recommendations below. Thank you again, for your critique of our manuscript.

**REVIEWER 2:**
As I mentioned in my public review, long-term treatment data would be especially helpful. Examining changes in the expression level of major Gal4 lines is also informative.

Please see our responses to Points #1 and #2 to Reviewer 1 in the “Public Reviews” section. Although examination of Gal4 expression patterns is extremely important, we believe that these analyses should be carefully performed on a case-by-case basis in the future for labs who wish to continue to use this methodology.

**REVIEWER 4:**
I feel addressing #2 would be a great addition to the current version, while #1 and #3 could be addressed in future studies or by researchers who are interested in these processes.Recommendation 1: “Both the metabolomics and transcriptome analyses were done using the whole animals, would it be more informative if these were done using specific tissue/organs such as the adult adipose tissue?”

Please see our response to Points #1 and #6 to Reviewer 1 in the “Public Reviews” section.

Recommendation 2: “Another related question is whether these detected changes are reversible or not after exposure to auxin? This would be informative for researchers to better design their temporally controlled experiments.”

We thank the reviewer for this suggestion and the analysis for this experiment is now provided in Figure S4I.

Recommendation 3: “Is spermatogenesis affected at all?”

We respectfully point out that many processes in spermatogenesis (as well as other biological processes) are affected by feeding (e.g., starvation) and would be extremely time consuming to carefully perform the analyses with the rigor required. We agree with Reviewer 4 and believe that this would be best to be performed on a case-by-case examination in the future.